# Electromagnetic reprogrammable coding-metasurface holograms

Lianlin Li[1], Tie Jun Cui[2,3], Wei Ji[1], Shuo Liu[2,3], Jun Ding [4], Xiang Wan[2,3],
Yun Bo Li[2,3], Menghua Jiang[4], Cheng-Wei Qiu[5,6,7] & Shuang Zhang[8]

Metasurfaces have enabled a plethora of emerging functions within an ultrathin dimension, paving way towards flat and highly integrated photonic devices. Despite the rapid progress in this area, simultaneous realization of reconfigurability, high efficiency, and full control over the phase and amplitude of scattered light is posing a great challenge. Here, we try to tackle this challenge by introducing the concept of a reprogrammable hologram based on 1-bit coding metasurfaces. The state of each unit cell of the coding metasurface can be switched between '1' and '0' by electrically controlling the loaded diodes. Our proof-of-concept experiments show that multiple desired holographic images can be realized in real time with only a single coding metasurface. The proposed reprogrammable hologram may be a key in enabling future intelligent devices with reconfigurable and programmable functionalities that may lead to advances in a variety of applications such as microscopy, display, security, data storage, and information processing.

[1] School of Electronic Engineering and Computer Sciences, Peking University, Beijing 100871, China. [2] State Key Laboratory of Millimeter Waves, Southeast University, Nanjing 210096, China. [3] Synergetic Innovation Center of Wireless Communication Technology, Nanjing 210096, China. [4] ECE Department, University of Massachusetts Lowell, Lowell, Massachusetts 01854, USA. [5] Department of Electrical and Computer Engineering, National University of Singapore, Engineering Drive 3, Singapore 117583, Singapore. [6] NUS Suzhou Research Institute (NUSRI), Suzhou Industrial Park, Suzhou 215123, China. [7] SZU-NUS Collaborative Innovation Center for Optoelectronic Science & Technology, Shenzhen University, Shenzhen 518060, China. [8] School of Physics and Astronomy, University of Birmingham, Birmingham B15 2TT, UK. Lianlin Li, Tie Jun Cui and Wei Ji contributed equally to this work. Correspondence and requests for materials should be addressed to L.L. (email: lianlin.li@pku.edu.cn) or to T.J C. (email: tjcui@seu.edu.cn) or to S.Z. (email: s.zhang@bham.ac.uk)

Holography is one of the most promising imaging techniques for recording the amplitude and phase information of light to reconstruct the image of objects[1]. Computer-generated holograms[2] are diffractive optical elements that encode the holographic information of object pattern calculated by computer. As a revolutionary three-dimensional imaging technique holography has attracted tremendous attention due to various applications in display, security, data storage, etc. However, most of the conventional holograms have either low resolutions or limited image quality. Their phase modulations rely on the light propagation over long distances inside the material in order to achieve the desired phase accumulation for wavefront shaping, resulting in large size of unit cells and large thickness of the phase holograms comparable to the wavelength.

Metasurfaces, ultrathin planar surfaces made of subwavelength metallic or dielectric elements, have shown the capability to arbitrarily manipulate the propagation and scattering of electromagnetic (EM) waves. Metasurfaces have emerged as promising alternatives for shaping wavefronts of light, and have recently been designed to realize versatile functionalities, including ultrathin planar lenses[3–6], optical vortex beam generations[7, 8], spin Hall effects of lights[9, 10], and metasurface holograms.

Compared with the conventional holograms, metasurface holograms have featured two major advantages. First, they provide unprecedented spatial resolution, low noise, and high precision of the reconstructed images, since both phase and amplitude information of the wavefront can be recorded in ultrathin and ultra-small holograms. Second, the pixel size is in the subwavelength scale that results in much higher transmission or reflection efficiency (the fraction of the transmitted or reflected energy of the light that contributes to the holographic image) than the traditional holograms due to the suppression or elimination of undesired diffraction orders. A number of metasurface holograms have been proposed and experimentally demonstrated in the terahertz, infrared, and visible regimes[11–21, 36, 41, 42] to achieve holographic images of high efficiency, good image quality, and full colour. Although metasurface holograms offer extraordinary advantages, only static holographic images

have been reported thus far because the phase/magnitude profiles are fixed once the metasurfaces are fabricated. Recently, dynamic/active metasurfaces or metamaterials by exploiting tunable or switchable materials have been proposed to realize various functionality, such as thermal-sensitive phase change materials $Ge_2Sb_2Te_5$[22, 23] for super-oscillation focusing[20] a vanadium dioxide[24–26] for beam scanning[21], applied-voltage sensitive graphene[27–29] for beam scanning[25, 26], mechanical actuation[30, 31] to reorient/rearrange the meta-atoms, and coherent controls of the light–matter interaction for all-optical logical operation and image processing[32]. Additionally, active elements (e.g., varactors and diodes) have been utilized on metasurfaces to demonstrate dynamic EM wave controls in microwave frequencies, such as the beam forming[33] and computational microwave imaging[34]. However, achieving dynamic hologram remains a challenging problem, and this holy grail is still far from being well addressed or solved so as to realize ultrathin, real-time pixel-level reconfigurable, and arbitrary holography. Metasurface-based holograms, with the judicious designs and pixel-level independent control of active components, may empower such ultimate dynamic capabilities for various applications that may revolutionarily advance the computational imager[35, 36], the wireless communication, and reproducing the digital EM environments.

In this article, we propose reprogrammable holograms by using 1-bit coding metasurfaces that are confirmed by proof-of-concept experiments in the microwave frequencies. By incorporating an electric diode into the unit cell of the metasurface, the scattering state of each individual unit cell can be controlled by applying different biased voltages across the diode. Thus, a single metasurface can accomplish various functions dynamically via the field programmable gate array (FPGA). To the best of our knowledge, this is the first reprogrammable hologram that can provide switchable phase modulations at microwave frequencies, with different high-resolution and low-noise holographic images generated by a single hologram. Our work addresses several critical issues typically associated with the current static metasurface holograms, featuring simplicity, being rewritable, high image quality, and high efficiency.

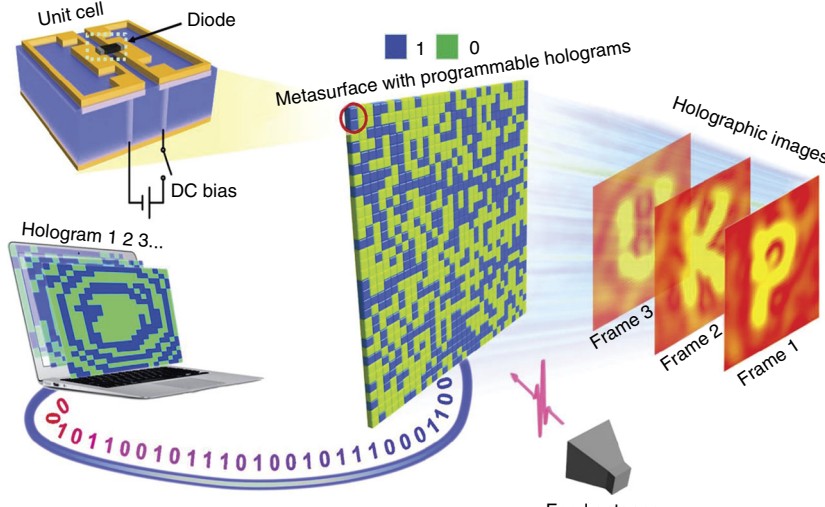

**Fig. 1** A sketch of the proposed dynamic holographic imaging. The metasurface in the middle is formed by an array of meta-atoms, with each having a pin diode welded between the two metallic loops and independently controlled by a DC voltage through a via (see the unit cell in the *upper left corner*). A computer digitally controls the metasurface by dynamically changing the phase distribution (Hologram 1, 2, 3,...) computed from the modified GS algorithm. Under the illumination of a feeding antenna (on the *bottom right side*), the metasurface hologram can successively project the holographic images (Frame 1, 2, 3...) at the imaging plane ($Z_r$), showing the letters 'P', 'K', 'U'

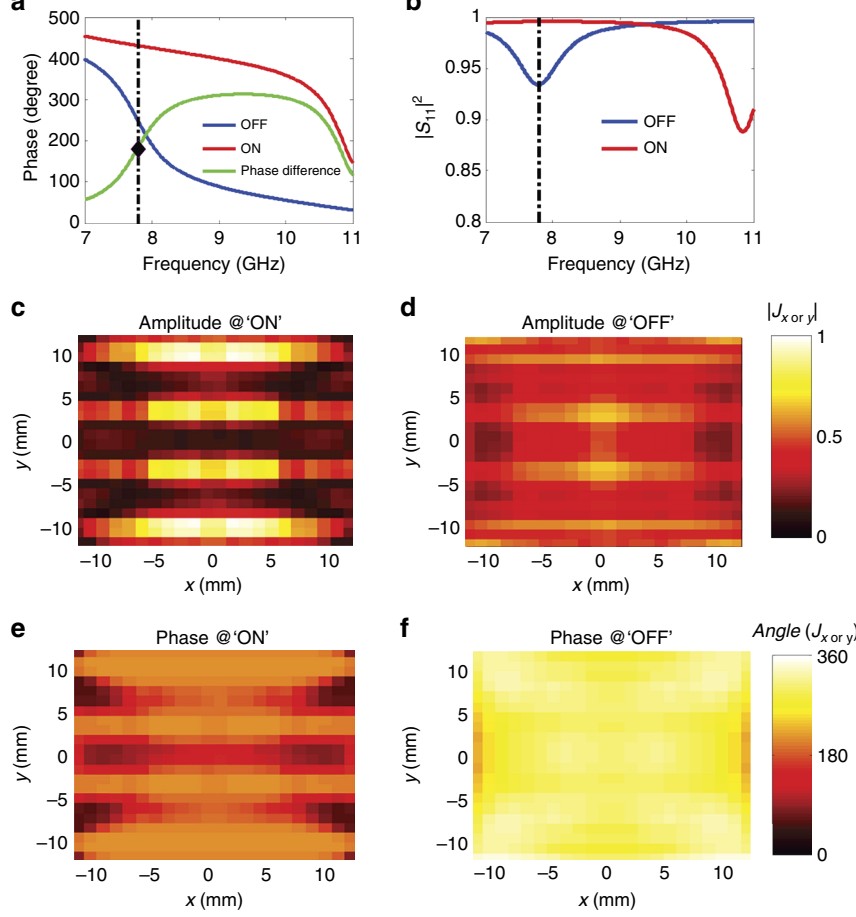

**Fig. 2** Phase and amplitude of the metasurface supercell. **a** The phase responses of the metamaterial supercell as the biased diode is 'OFF' and 'ON' over a range of frequencies. **b** The reflection efficiency of the metaatom when the biased diode is at the state of 'OFF' and 'ON'. **c–f** The distributions of normalized amplitudes and phases of the equivalent currents on the half-wavelength supercell as the biased diode is at 'OFF' and 'ON' states. The equivalent current is retrieved at the frequency of 7.8 GHz by performing the source inversion technique

## Results

**Design of reprogrammable hologram.** The conventional Gerchberg–Saxton (GS) algorithm[36] has been modified for generating the binary phase profile of the reprogrammable hologram (see Methods). The effective induced current is first introduced to characterize the realistic EM response of the metasurface unit cells in order to address the accuracy issue arising from the point-like dipole model. The induced current can be obtained by applying the Huygens's principle or the source inversion technique[37]. Each unit cell of the dynamic metasurface assumes one of the two switchable states: '0' and '1', while the induced effective current associated with each state solely determines the radiation property of the unit cell. Consequently, the design of the reprogrammable metasurface hologram is mathematically casted into a combinatorial optimization problem, as opposed to the continuous-valued optimization problem solved in the conventional GS algorithm. To solve this combinatorial optimization problem, the standard GS algorithm is reformulated, and more careful considerations have been given to the back/forward propagation operations, as illustrated in Methods.

Figure 1 illustrates a sketch of the proposed dynamic holograph imaging with a 1-bit reprogrammable metasurface. The meta-atom (unit cell) of the 1-bit reprogrammable metasurface is parametrically tailored for our purpose based on our previously developed metamaterial[33]. In each unit cell, two planar symmetrically patterned metallic structures are printed on the top surface of the F4B substrate with a dielectric constant of 2.65 and a loss tangent of 0.001, with a diode loaded between them. For each planar structure, four metallic via holes are drilled through the substrate to connect the top structure with four pieces of separated grounds on the bottom surface of the substrate that are used to apply the direct current (DC) bias voltage. Thus, each unit cell can be independently programmed to realize the required '0' or '1' state by controlling the applied voltage bias of the diode. The unit cell size is $6 \times 6 \times 2 \ mm^3$ that is $\sim 0.156 \times 0.156 \times 0.05 \ \lambda^3$ at the frequency of 7.8 GHz (see Supplementary Fig. 1a in Supplementary Note 1). From simulations it is found that the scattering of the metamaterial particle is dominated by the corners of the unit cell. Hence, the near-field coupling is substantially strong if the corners of the two nearby cells are close enough to each other. To suppress the corner-related scattering effect, $5 \times 5$ unit cells are grouped to form a half-wavelength supercell.

The diode is at the state of 'ON' (or 'OFF') with a biased voltage of 3.3 V (or 0 V), and the corresponding equivalent circuit is illustrated in the upper (lower) part of Supplementary Fig. 1b. The numerical results obtained by inserting the circuit models into the CST Microwave Studio show that the metamaterial particle behaves as a '1' ('0') element when the diode is 'ON' ('OFF'). As shown in Fig. 2a, the transmission phase difference between the '1' and '0' states obtained by the CST S-parameter simulation is ∼180° around the frequency of 7.8 GHz. Figure 2b shows that the theoretical reflection efficiency can reach above

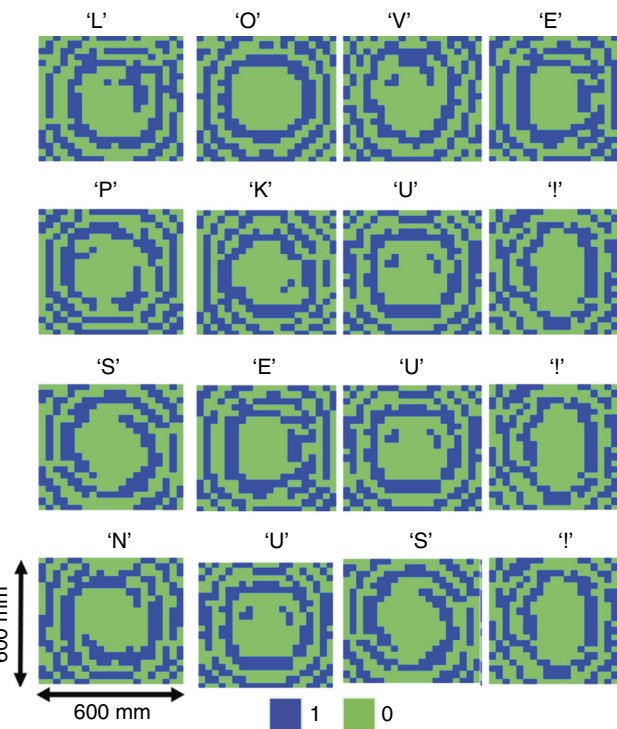

**Fig. 3** The binary phase profiles of the coding metasurface for different holographic images. The binary holograms for a sequence of letters of 'LOVE PKU! SEU! NUS!' are generated by the modified GS algorithm that could be realized by switching the states of corresponding unit cells of the coding metasurface

90%. Figure 2c–f compares the amplitude and phase distributions of the current induced on the half-wavelength supercell at the '0' and '1' states at 7.8 GHz, respectively. It can be seen that the induced current is almost homogeneously distributed across the supercell. One interesting observation is that when the state of the metasurface supercell is switched from '1' to '0' (or from '0' to '1'), the amplitudes of the induced current remain almost unchanged, while the phases are flipped by ~180°.

**Experimental results**. The reprogrammable hologram is validated by proof-of-concept experiments at microwave frequencies. To confirm the reconfigurable ability of the metasurface, a sequence of holographic images, namely a sentence of 'LOVE PKU! SEU! NUS!', are generated. The 1-bit coding metasurface is composed of $20 \times 20$ supercells, covering an area of $600 \times 600$ mm$^2$. The dynamic metasurface hologram is fabricated by using the standard printed circuit board process on a FR4 substrate ($\varepsilon_r \approx 2.65$). By toggling different applied voltages to control the 'ON' and 'OFF' states of the diodes, the required distribution of '0' and '1' elements across the dynamic metasurface hologram can be achieved. In the experiment, a feeding antenna with working bandwidth from 6 to 14 GHz is employed to generate the $x$-polarized quasiplane waves for illuminating the coding metasurface, and a standard waveguide probe is used to scan the image plane with a resolution of $5 \times 5$ mm$^2$ to obtain the holographic images(see more details in Methods).

The phase profiles of the holograms for the letters obtained by the modified GS algorithm are plotted in Fig. 3, in which the green and blue colours represent two different digital states with constant amplitude. The corresponding simulation results by commercial software, CST Microwave Studio, are plotted in Fig. 4. In the numerical simulations, the holograms are illuminated by

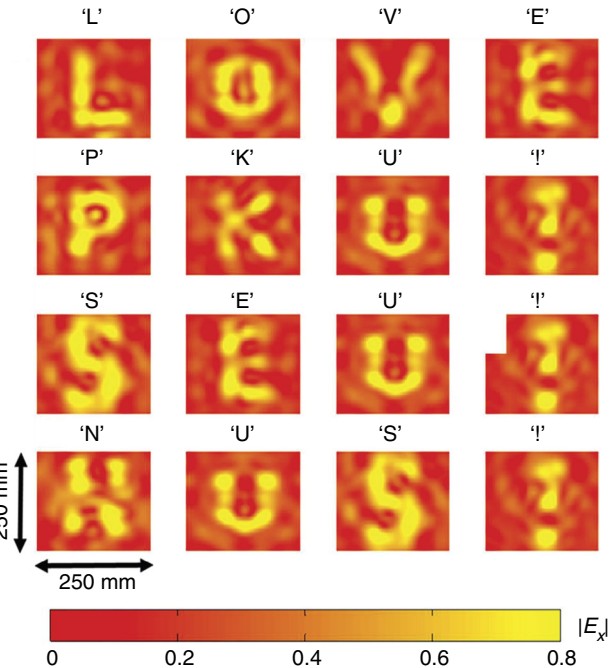

**Fig. 4** The simulation results of holographic images with the phase profiles in Fig. 3. The simulated holographic images ($E_x$-field intensity) of 'LOVE PKU! SEU! NUS!' corresponding to the holograms shown in Fig. 3 are obtained at the image plane of $Z_r = 0.4$ m with an $x$-polarized incidence

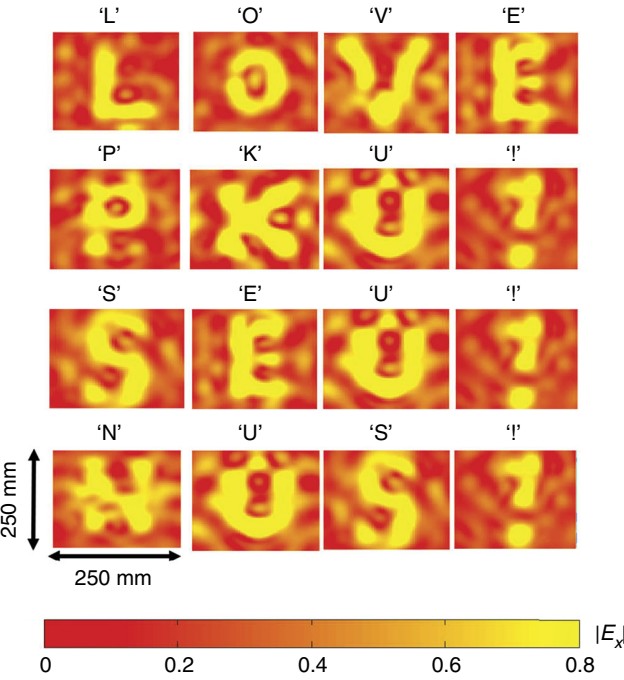

**Fig. 5** The experimental results of holograph images with the phase profiles in Fig. 3. The experimentally observed holographic images ($E_x$-field intensity) of 'LOVE PKU! SEU! NUS!' corresponding to the holograms shown in Fig. 3 are measured at the image plane of $Z_r = 0.4$ m, that agree well with the simulation results illustrated in Fig. 4

an $x$-polarized plane wave, and the holographic image is detected at $Z_r = 400$ mm that clearly shows the letters 'LOVE PKU! SEU! NUS!'. The experimentally observed holographic images of 'LOVE PKU! SEU! NUS!' by setting the corresponding phase distribution across the coding metasurface is presented in Fig. 5,

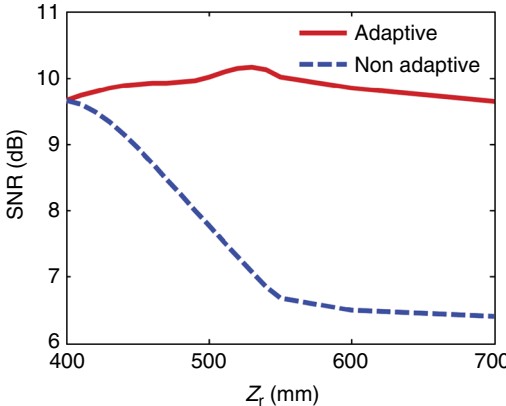

**Fig. 6** The dependence of SNR for the holographic image as the function of observation distance. The *red line* represents SNR for the holographic image as a function of the observation distance with the adaptive adjustment of the metasurface hologram, while the *blue dashed line* denotes SNR for the holograph image with the observation distance fixed at $Z_r = 400$ mm. These two lines are obtained by averaging the results of 'LOVE PKU! SEU! NUS!'

which is in very good agreement with the CST-based numerical simulations shown in Fig. 4. An overall efficiency of ~60% (defined as the fraction of the incident energy that contributes to the holographic image) is achieved in our metasurface hologram design that is lower than the simulated values (above ~90%) due to the phase-quantization loss[38] (see Supplementary Fig. 5), the non-ideal plane-wave illumination, the imperfect quality of commercial PIN diode, etc., as discussed in Supplementary Note 5. We also achieved a signal-to-noise ratio (SNR) of ~10 in the metasurface hologram sample that is defined as the ratio of the peak intensity in the image to the standard deviation of the background noise[19].

## Discussion

The hologram optimized by the modified GS algorithm strongly depends on the position of the image plane. To prove the robustness of the metasurface hologram against the observation distance, Supplementary Note 3 presents the evolution of the holographic images of 'S' with the increase of observation distance, i.e., 400, 440, 480, and 700 mm, where the optimal hologram is achieved at $Z_r = 400$ mm. It is noticed that the quality of the holographic image is acceptable when the observation distance is <500 mm. Hence, the hologram requires adjustment to achieve clearer holographic images when the holographic plane is relocated. Consequently, the quality of holographic image can be dramatically improved by adaptively reprogramming the metasurface hologram for the observation distance of interest. Supplementary Movie 1 shows the evolution of the holographic images with increasing observation distance for the metasurface hologram with and without being adaptively tuned, clearly demonstrating that high-quality image can be maintained for different image locations with adaptive tuning. Figure 6 presents a quantitative analysis in terms of SNR of the holographic images, from which we clearly see that the image quality is significantly improved by 4–5 dB when the hologram is adaptively tuned. Apparently, the reprogrammable metasurface hologram can provide a higher-quality image in a more flexible way. Although the reprogrammable metasurface hologram is designed at 7.8 GHz, it works well within an operating frequency bandwidth of 0.5 GHz (see Supplementary Figs 3 and 4 in Supplementary Note 4). Similarly, it is expected that the holographic image

quality can be improved when the operational frequency changes if we adaptively adjust the hologram.

Here are programmable 1-bit coding metasurface hologram has been proposed and experimentally realized that can, in principle, generate arbitrary holographic images by programming the input control voltages. A prototype of the reprogrammable metasurface hologram was fabricated to produce different images in the microwave frequencies. The proof-of-concept experimental results agree very well with the numerical simulations, validating the programmable metasurface as a viable means for creating more sophisticated holograms. In the experiments, the switching speed of the PIN diode is 3 ns, and hence the total hologram reconfiguration time depends mainly on the configuration of the control circuit. When the PIN diodes are controlled in parallel by FPGA, the reconfiguration time is completely determined by the FPGA clock (CLK) rate. On the other hand, if the PIN diodes are controlled in the clustered way in case the number of FPGA Input/Output (I/O) is not enough, the reconfiguration time would be increased by a factor of the cluster number. In our experiments, we adopt the parallel way, in which the FPGA clock rate is 100 MHz, corresponding to the time of each operation cycle of 10 ns. To change the PIN diode status in parallel, three operation cycles are needed after compiling. Thus, the total hologram reconfiguration time is ~33 ns. Although the proposed metasurface hologram works in the mode of 1-bit coding and phase-only modulation, the reprogrammable metasurface holograms can be readily extended to exhibit multiple bits and both phase and amplitude modulations that can lead to more advanced, efficient, and versatile devices with adaptive and rewritable functionalities.

We remark that the reconfigurable metasurface hologram may also be realized at higher frequencies using the proposed methodology by constructing 1-bit phase-modulated light spatial masks. Several well-established switchable diodes may facilitate the frequency scaling, for instance, the Schottky diode at the terahertz[39, 40] and the thermal $VO_2$ diode in the infrared-and-visible frequencies[41, 42]. Additionally, by incorporating other phase changing materials like $Ge_2Sb_2Te_5$[22, 23] into appropriate nano-antenna design, it is possible to extend our concept of dynamic metasurface holography to higher frequencies through dynamically modifying the phase of scattering from each individual antennas, while maintaining the same scattering magnitude.

## Methods

**The modified GS algorithm.** As shown in Eq. 1, the hologram **E** is linearly related to the current **J** induced over the metasurface illuminated by a plane wave. However, the effective induced current **J** is dependent on the binary metasurface hologram **S** in a nonlinear manner. For the purpose of numerical computational implementation, Supplementary Eq. 1 is reformulated in the compact form, i.e., $\mathbf{E} = \mathbf{AJ}$, where the mapping matrix with entries of $\int_\Delta g(\mathbf{r}, \mathbf{r}_{n_x,n_y} + \delta\mathbf{r})\mathbf{J}^{(S_{n_x,n_y})}(\delta\mathbf{r})\mathrm{d}\delta\mathbf{r}$ comes from Supplementary Eq. 1. Similar to the conventional GS algorithm, the modified GS algorithm works in the manner of alternating projections. It attempts to minimize the nonconvex objective function defined by

$\sum_{i=1}^{N_x N_y} \left( E^{\mathrm{des}}(i) - |E(i)| \right)^2$, where $E^{\mathrm{des}}(i)$ denotes the value of desired image at the $i$th pixel. The algorithm starts with a random initialization and iteratively imposing the hologram domain (support) and measured amplitude constraints using projections, as summarized in Table 1. The process converges after several iterations. Supplementary Movie 2 is provided to illustrate the behaviour of images with the progress of iterations. The source inversion technique is employed to calibrate the current distributions of $\mathbf{J}^{(1)}$ and $\mathbf{J}^{(0)}$. After that, the metasurface particle either at the state of '1' or '0' is illuminated by a plane wave, and the resultant copolarized electric field is collected at one wavelength away from the metasurface particle. The calibration measurements are organized into a column vector **e**. To characterize the inhomogeneous current induced within the metasurface particle, one metasurface particle is regarded as an array of $10 \times 10$ point-like dipoles, stacked in a 100-length column vector of **p**. Apparently, the calibration vector **e** is linearly related to the vector **p** as $\mathbf{e} = \mathbf{Gp}$, where the entries of **G** come from the three-dimensional

---

**Table 1 The modified GS algorithm**

Initial solution: $\mathbf{E} = |\mathbf{E}^{des}|$
WHILE until when no further improvement can be obtained if more iterations are applied.
$\mathbf{J} = \mathbf{A}^{\dagger}\mathbf{E}$
 FOR $n_x = 1 : N_x$
 FOR $n_y = 1 : N_y$
 $\mathbf{J}^{(s_{n_x,n_y})} = \begin{cases} \mathbf{J}^{(1)}, \text{if} & -\pi/2 \leq \text{angle}(J(n_x,n_y)) \leq \pi/2; \\ \mathbf{J}^{(0)} & \text{otherwise} \end{cases}$
 END FOR
 END FOR
$\mathbf{E} = \mathbf{A}\mathbf{J}$
$\mathbf{E} = |\mathbf{E}^{des}| \odot \exp(j \cdot \text{angle}(\mathbf{E}))$, Herein, $\odot$ denotes the element-wise product.
ENDWHILE

---

Green's function in free space. Then, the complex-valued amplitude of **p** can be readily retrieved by solving the least-square problem of $\mathbf{e} = \mathbf{G}\mathbf{p}$.

**Measurement system**. To complete the measurement calibration and measure the normalized holographic image, an experimental setup is established in an anechoic chamber with the size of $2 \times 2 \times 2\,m^3$ that includes a transmitting (Tx) horn antenna, a waveguide probe as a receiver, and a vector network analyzer (VNA, Agilent E5071C), as illuminated in Supplementary Fig. 4. In our imaging experiments, the VNA is used to acquire the response data by measuring the transmission coefficients ($S_{21}$). More specifically, the horn antenna and waveguide tip are connected to two ports of a VNA through two 4-m-long 50 Ω coaxial cables. To suppress the measurement noise level, the average number and filter bandwidth in the VNA are set to 10 and 10 kHz, respectively. Both the feeding antenna and sample were coaxially mounted on a board at a distance of 1.8 m that was chosen based on the consideration that the feeding antenna should be placed inside the Fraunhofer region of the coding metasurface. The Fraunhofer distance can be calculated by $R = 2L^2/\lambda$, where $L$ is the maximum electric length of the antenna and $\lambda$ the working wavelength. This distance ensures that the optical path difference between the centre and the edge of the coding metasurface is less than $\lambda/16$, providing a quasiplane wave incidence as considered in the far-field numerical simulations. As $R$ is proportional to $L^2$, the distance between the feeding antenna and coding metasurface becomes too large for our microwave chamber, and this is the reason why the size of the fabricated sample is reduced to half of the model in simulations. One may notice that the distance 1.8 m in the experiment is smaller than the minimum distance $R$ calculated from the formula. We remark that the aforementioned distance of the far-field region was obtained from the assumption of point source excitation, whereas in the experimental case, the incident wavefront generated from the horn antenna is relatively flat, and the corresponding distance $R$ could be reduced accordingly.

**Data availability**. The authors declare that all relevant data are available in the paper and its Supplementary Information Files, or from the corresponding author on request.

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

## Acknowledgements

This work was supported in part from the National Natural Science Foundation of China under Grant Nos. 61471006, 61631007, and 61571117, and in part from the 111 Project under Grant No. 111-2-05. C.-W.Q. acknowledges the support from the National Research Foundation, Prime Minister's Office, Singapore, under its Competitive Research Programme (CRP Award No. NRF-CRP15-2015-03). S.Z. acknowledges financial support from ERC consolidator grant (TOPOLOGICAL), the Royal Society, the Wolfson Foundation, and Leverhulme (RPG-2012-674).

## Author contributions

L.L., T.J.C., C.-W.Q., and S.Z. conceived the idea. L.L. conducted the numerical simulations and theoretical analysis. L.L., T.J.C., C.-W.Q., and S.Z. wrote the manuscript. All authors participated in the experiments and data analysis and read the manuscript.

## Additional information

**Competing interests:** The authors declare no competing financial interests.

