## [Peer Review file · Nature Communications]

Reviewers' comments:

Reviewer #1 (Remarks to the Author):

The authors present a scheme for electrically reconfigurable metasurface at RF frequencies and use it to demonstrate the ability to produce varying holograms by reprogramming the phase response of the metasurface. This is being achieved by applying a DC voltage on a diode located in each of the individual antennas comprising the metasurface and changing their phase response between 0 and π . The authors refer to these phase shift as binary coding of the metasurface and present a modified version of the well-known GS algorithm which optimizes the binary coding of the metasurface in order to obtain the desired phase response.

This is an interesting study which as far as I know constitutes the first demonstration of electrically reconfigurable metasurface. The authors utilizes the results of their previous studies on controlling the phase response of antennas in the 6-12 GHz range for reducing the RCS of metallic surfaces by applying it to a new application. In my opinion, this study could be of interest to meta surfaces and metamaterials scientific community. However, I cannot recommend the publication of the manuscript in its current format as many details are missing as well as some perspective and context as detailed below. Nevertheless, if the authors address properly these issues I believe the manuscript will be suitable for publication in Nature Communications.

1. The authors claim that they are the first to present a reconfigurable metasurface. While this might be correct when limited to the RF and for electrical tuning, reconfigurable metasurfaces in optical frequencies (though utilizing different mechanisms) have been studied and demonstrated. These studies, listed below, should be cited and briefly discussed in order to put the work presented by the authors in the right perspective and context. Nano Lett. 11, 2142 (2011); Nature Nanotech. 8, 252 (2013); Nat. Nano. 6, 630 (2011); Appl. Phys. Lett. 103, 104103 (2013); Nano Lett. 13, 1257 (2013); Phys. Rev. Lett. 109, 083902 (2012); Phys. Rev. Lett. 103, 147401 (2009); Opt. Mater. Express 5, 2513 (2014); Adv. Mater. 25, 3050 (2013); Light Sci. Appl. 5:e16070 (2016); Nat. Photon. 10, 60 (2016).
2. The authors indicate that they present a "proof of concept" for reconfigurable metasurfaces for holography applications which is demonstrated in the RF regime. Nevertheless, the more interesting applications for holography, particularly reconfigurable holography, are located at much higher frequency (IR and visible) – regimes in which the demonstrated scheme is less applicable. The authors should address this point by elaborating on the upper spectral limit at which their scheme can operate as well as elaborating on their ideas of how to realize such metasurfaces at optical frequencies.
3. Efficiency. The theoretical analysis of the authors predicts overall efficiency of $\sim 90\%$ while the experiments reveal a substantially lower overall efficiency of $\sim 60\%$. I find this difference rather surprising as the properties of the materials at RF frequencies are well known and the expected variations and errors in the dimensions of the structures are expected to be quite small (compared e.g. to nanostructure at optical frequencies). In fact, metasurfaces holography at optical frequencies, which is presumably more difficult, was able to demonstrate equivalent and even superior efficiency (Nano Lett. 14, 2485 (2014), Nat. Nano 10, 296 (2015)). The authors should explain this large deviation between experiments and theory and identify the main mechanisms limiting the efficiency of their metasurfaces.
4. The authors use a 5x5 array of identical antennas in order to construct a single pixel of their metasurface. Although the authors do not say that explicitly, this technique reduces the impact of coupling between neighboring antennas and provides an overall pixel phase response which is more similar to that of the individual antennas (or that of an infinite periodic array). Consequently, this approach reduces the resolution to approximately half wavelength which is similar to that of conventional holography. The authors should state that and elaborate on their ideas (if any) of obtaining real sub-wavelength resolution (which in this case could be $\sim \lambda/10$).
5. What is the hologram reconfiguration time? Can one obtain high frame rate using the presented approach? What is the frame rate limit?
6. The authors indicate that the bandwidth of their hologram is $\sim 0.5\text{GHz}$, and show SNR measurements in order to support that claim. What they should do is present complete images of

the obtained image at several points within that band to prove that there is no deterioration in the image quality.

7. The authors utilize a single bit coding scheme in order to design and realize the metasurfaces. However, based on the data shown in Fig. 2(a) one may expect that the phase response between the two states should be continuously tunable. Obtaining an analog phase response (or at least a several bits coding scheme) can improve substantially the quality of the obtained images in terms of resolution, distance dependence and probably efficiency. I find it peculiar that the authors have not pursued this option in this paper and I expect them to explain why they have chosen to limit themselves to a 1 bit scheme and if there is an option to have a continuous phase tuning using their system.

8. What is the stop criterion the authors use in their design loop (Table 1)?

9. Please add dimensions/axes to Fig. 2(c-f).

Reviewer #2 (Remarks to the Author):

In this paper, authors experimentally demonstrate reconfigurable holograms at Microwave frequency. Demonstrated hologram is 2-level amplitude hologram. Switching scheme is obtained by introducing a diode between two adjacent antenna cells, and biasing it using a DC bias voltage. The hologram is observed by sweeping a waveguide probe and measuring the wave amplitude at a 2D plane away from the microwave antenna array. The simplistic approach is novel and demonstrates a switchable hologram in the microwave region.

1. The word "metasurface" usually refer to subwavelength periodicity. The paper shows the unit cell to be 6mmx6mm, and operating frequency band to be 7.8GHz (wavelength 3.8mm). To call the antenna array a metasurface is perhaps misleading in this context.

2. At present context, the title of the paper is too broad and perhaps should specify that the holograms demonstrated are "Microwave holograms".

3. The authors should comment on the switching speed of the hologram, from one state to another.

4. Authors should comment if their demonstration scheme is scalable for optical frequencies.

5. The paper mentions no applications of the demonstrated microwave hologram. It should elaborate further on how microwave holograms could be useful.

The paper should be rewritten to focus on the achieved results. The results are indeed publishable. But, at its present form, I cannot recommend but to recommend rejection of the paper from Nature Communications.

Re: Manuscript # NCOMMS-16-28675 “Reprogrammable coding-metasurface holograms in the microwave regime,” by L. Li et al.

Responses to Reviewers' Comments

We appreciate very much the constructive and valuable comments from both reviewers, which have really helped improve the quality of the manuscript. Based on such comments and suggestions, we have revised the manuscript very carefully and thoroughly. The explanations and corrections to all comments were added in the revised manuscript, as marked by the yellow background. Below are our item-by-item responses (in blue fonts) to the comments (in black fonts).

To Reviewer #1

Q1. The authors claim that they are the first to present a reconfigurable metasurface. While this might be correct when limited to the RF and for electrical tuning, reconfigurable metasurfaces in optical frequencies (though utilizing different mechanisms) have been studied and demonstrated. These studies, listed below, should be cited and briefly discussed in order to put the work presented by the authors in the right perspective and context. Nano Lett. 11, 2142 (2011) ; Nature Nanotech. 8, 252 (2013) ; Nat. Nano. 6, 630 (2011); Appl. Phys. Lett. 103, 104103 (2013); Nano Lett. 13, 1257 (2013) ; Phys. Rev. Lett. 109, 083902 (2012); Phys. Rev. Lett. 103, 147401 (2009) ; Opt. Mater. Express 5, 2513 (2015) ; Adv. Mater. 25, 3050 (2013) ; Light Sci. Appl. 5:e16070 (2016) ; Nat. Photon. 10, 60 (2016) .

Our Response:

Thank you very much for bringing these related references into our attention. These works have been cited and briefly discussed in the revised manuscript. For details, please refer to the second paragraph at Page 2 in the revised manuscript (also shown below).

“Recently, dynamic/active metasurfaces or metamaterials by exploiting tunable or switchable materials have been proposed to realize various functionality, such as thermal-sensitive phase change materials $\text{Ge}_2\text{Sb}_2\text{Te}_5$ ²¹⁻²² for super-oscillation focusing, vanadium dioxide for beam scanning, applied-voltage sensitive graphene²⁶⁻²⁸ for beam scanning, mechanical actuation^{29,30} to reorient/rearrange the meta-atoms, and coherent controls of the light-matter interaction for all-optical logical operation and image processing³¹. Additionally, active elements (e.g., varactors and diodes) have been utilized on metasurfaces to demonstrate dynamic EM wave controls in microwave frequencies, such as the beam forming³² and computational microwave imaging³³. However, the dynamic hologram remains a challenging problem, and this holy grail is still far from being well addressed or solved so as to realize ultrathin, real-time pixel-level reconfigurable, and arbitrary holography. Metasurface-based holograms, with the judicious designs and pixel-level independent control of active components, may empower such ultimate dynamic capabilities for various applications which may revolutionarily advance the computational imager^{33,34}, the wireless communication, and reproducing the digital EM environments.”

Q2. The authors indicate that they present a “proof of concept” for reconfigurable metasurfaces for holography applications which is demonstrated in the RF regime. Nevertheless, the more interesting applications for holography, particularly reconfigurable holography, are located at much higher frequency (IR and visible) – regimes in which the demonstrated scheme is less applicable. The authors should address this point by elaborating on the upper spectral limit at which their scheme can operate as well as elaborating on their ideas of how to realize such metasurfaces at optical frequencies.

Our Response:

Thank you very much for the suggestions. In fact, the reconfigurable metasurface hologram may also be realized at higher frequencies using the proposed methodology via the construction of 1-bit phase-modulated light spatial mask. We note that several well-established switchable metamaterials may be used for this purpose, for instance, VO₂ at the terahertz [1] and near-infrared [2] frequencies, and $\text{Ge}_2\text{Sb}_2\text{Te}_5$ at the near-infrared frequency [3-4], etc. By incorporating phase changing materials into appropriate nanoantenna design, it is possible to extend our concept of dynamic metasurface holography to higher frequencies through dynamically modifying the phase of scattering from each individual antennas, while maintaining the same scattering magnitude.

[1] D. Wang *et al.*, “Switchable ultrathin quarter-wave plate in terahertz using active phase-change metasurface,” *Sci. Rep.* 5, 15020, Oct. 2015.

[2] M. J. Dicken *et al.*, “Frequency tunable near-infrared metamaterials based on VO₂ phase transition,” *Opt. Express* 17, 18330, Sep. 2009.

[3] B. Gholipour, J. Zhang, K. F. MacDonald, D. W. Hewak, and N. I. Zheludev, “An all-optical, non-volatile, bidirectional, phase-change meta-switch,” *Adv. Mater.* 25, 3050-3054, Jun. 2013.

[4] Q. Wang *et al.*, “Optically reconfigurable metasurfaces and photonic devices based on phase change materials,” *Nat. Photonics* 10, 60–65, Dec. 2015.

This has been addressed in the Conclusion in the revised manuscript.

Q3. Efficiency. The theoretical analysis of the authors predicts overall efficiency of ~90% while the experiments reveal a substantially lower overall efficiency of ~60%. I find this difference rather surprising as the properties of the materials at RF frequencies are well known and the expected variations and errors in the dimensions of the structures are expected to be quite small (compared e.g. to nanostructure at optical frequencies). In fact, metasurfaces holography at optical frequencies, which is presumably more difficult, was able to demonstrate equivalent and even superior efficiency (Nano Lett. 14, 2485 (2014), Nat. Nano 10, 296 (2015)). The authors should explain this large deviation between experiments and theory and identify the main mechanisms limiting the efficiency of their metasurfaces.

Our Response:

We thank the reviewer for pointing out the discrepancy of efficiency between the measurement and theory. There are a few major factors to cause this relatively low conversion efficiency (~60%) as follows:

- There is the quantization loss due to the phase level, as pointed out in Ref. [1];
- In our experiments, the metasurface with finite extension is illuminated by a horn antenna, therefore there exists finite divergence angle in the incident beam. However, the simulation results are obtained when the metasurface atom is illuminated by an idea plane wave in the periodic boundary condition.
- The pin-diode is a commercial product purchased from commercial supplier, which are not ideal as the perfectly lossless and stable diode used in the theory. This imperfection in the product quality will be more exaggerated in our ambitious aim of achieving “reconfigurable”/“arbitrary” holography, since each pixel needs to be individually and independently controlled via diode. That differs from those papers mentioned before, since those reported works are mostly to realize tenability via controlling the overall piece of phase change materials at the same pace, or some recent works with pixel-level control via focused laser pulse writing GST attempted only one functionality of focusing, rather than sophisticated holography. In addition, once the focusing depth needs to change, one needs to restore the GST sample and rewrite a new pattern. In our case, our “motherboard” layout structure has no change, while the big quantity of pin-diode and the limited phase

steps will add up to the loss. It is the price for the advanced versatility and reconfigurability, which don't come for free.

[1] Y. Yifat, M. Eitan, Z. Iluz, Y. Hanein, A. Boag, and J. Scheuer, "Highly efficient and broadband wide-angle holography using patch-dipole nanoantenna reflectarrays," *Nano Lett.* 14, 2485-2490, May 2014

Q4. The authors use a 5x5 array of identical antennas in order to construct a single pixel of their metasurface. Although the authors do not say that explicitly, this technique reduces the impact of coupling between neighboring antennas and provides an overall pixel phase response which is more similar to that of the individual antennas (or that of an infinite periodic array). Consequently, this approach reduces the resolution to approximately half wavelength which is similar to that of conventional holography. The authors should state that and elaborate on their ideas (if any) of obtaining real sub-wavelength resolution (which in this case could be $\sim\lambda/10$).

Our Response:

We appreciate very much this comment. As clearly pointed by the reviewer, the electromagnetic (EM) interaction between neighboring pixels cannot be neglected due to the close gap between the neighboring metasurface unit cells, which is necessary for enhancing the efficiency. As a consequence, we choose 5x5 array of identical unit cells as a pixel, which is roughly 80% of the free space wavelength. Based on the Nyquist-Shannon theorem, a pixel size of half wavelength would be sufficient to contain all the information for generating a broad-angle free space holographic image. The pixel size of our sample is close but not there yet. As a future work, we will look into optimisation of the metasurface, by adjusting the gap size between neighboring unit cells, the number of unit cells in a supercell, so as to reduce the size of the supercell to within half the wavelength, and meanwhile maintaining a reasonably low impact of nearest neighboring coupling.

Q5. What is the hologram reconfiguration time? Can one obtain high frame rate using the presented approach? What is the frame rate limit?

Our Response:

Thank you very much for this insightful comment. Below is our response to this comment, which is addressed in the revised manuscript (Page 4, the last paragraph):

The switching speed of the PIN diode is 3 ns, and hence the total hologram reconfiguration time depends mainly on the configuration of the control circuit. In our experiment, the PIN diodes are controlled in parallel by FPGA, the reconfiguration time is completely determined by the FPGA clock (CLK) rate, which is 100 MHz, corresponding to the time of each operation cycle of 10 ns. To change the PIN diode status in parallel, 3 operation cycles are needed after compiling. Thus the total hologram reconfiguration time is around 33ns. We are in the process of producing a new FPGA, whose CLK rate can reach $F_c = 320$ MHz or beyond.

Q6. The authors indicate that the bandwidth of their hologram is ~ 0.5 GHz, and show SNR measurements in order to support that claim. What they should do is present complete images of the obtained image at several points within that band to prove that there is no deterioration in the image quality.

Our Response:

Thank you very much for your good suggestion. Several images at different sampling frequencies are plotted in Figure S4 in the Supplementary Information, which is also provided as follows.

Supplementary Figure S4 | The image of a letter “P” at different sampling frequencies (the holographic images are achieved by adaptively tuning the metasurface holograms while changing the working frequencies).

Q7. The authors utilize a single bit coding scheme in order to design and realize the metasurfaces. However, based on the data shown in Fig. 2(a) one may expect that the phase response between the two states should be continuously tunable. Obtaining an analog phase response (or at least a several bits coding scheme) can improve substantially the quality of the obtained images in terms of resolution, distance dependence and probably efficiency. I find it peculiar that the authors have not pursued this option in this paper and I expect them to explain why they have chosen to limit themselves to a 1 bit scheme and if there is an option to have a continuous phase tuning using their system.

Our Response:

We thank the reviewer for this constructive comment. The diode response is highly nonlinear, and therefore it is highly challenging to precisely manipulate the intermediate states between 'ON' and 'OFF' at current stage. Hence, we resort to 1-bit scheme. Nevertheless, we leverage on the spatial degree of freedom and judiciously exploit the 1-bit coding at different pixels to still provide dynamic phase tuning.

Q8. What is the stop criterion the authors use in their design loop (Table 1)?

Our Response:

Thank you very much for this question. We stop the iteration when no further improvement can be obtained if more iterations are applied. Actually, only a few iterations are needed for achieving this stop criterion in our study cases. We add a video to illustrate the images with the progress of iterations in the Supplementary Information.

9. Please add dimensions/axes to Fig. 2(c-f).

Our Response:

We thank the reviewer for this suggestion. The dimensions and axes have been added in Figure 2(c-f) in the revised manuscript.

To Reviewer #2

In this paper, authors experimentally demonstrate reconfigurable holograms at Microwave frequency. Demonstrated hologram is 2-level amplitude hologram. Switching scheme is obtained by introducing a diode between two adjacent antenna cells, and biasing it using a DC bias voltage. The hologram is observed by sweeping a waveguide probe and measuring the wave amplitude at a 2D plane away from the microwave antenna array. The simplistic approach is novel and demonstrates a switchable hologram in the microwave region.

Q1. The word “metasurface” usually refer to subwavelength periodicity. The paper shows the unit cell to be 6mmx6mm, and operating frequency band to be 7.8GHz (wavelength 3.8mm). To call the antenna array a metasurface is perhaps misleading in this context.

Our Response:

Thank you very much to raise this question. At 7.8 GHz, the wavelength is 38.4 mm, but **not 3.8 mm**. Thus, the unit cell is about $\sim\lambda_0/6$, and we use the concept of metasurface to design this dynamic hologram.

2. At present context, the title of the paper is too broad and perhaps should specify that the holograms demonstrated are “Microwave holograms”.

Our Response:

Thank you very much for this good suggestion. The microwave demonstrations were proof-of-concept, and like many other papers about digital metamaterials or microwave metasurfaces, we didn't specify frequency band in the title while indeed indicating that clearly in the Abstract and main text. In this version, the title has been changed into “Electromagnetic Reprogrammable Coding-metasurface Holograms”, to reflect that the work is in the EM band.

3. The authors should comment on the switching speed of the hologram, from one state to another.

Our Response:

Thank you very much for this constructive suggestion. Below is our response to this comment, which is addressed in the revised manuscript (Page 4, the last paragraph):

The switching speed of the PIN diode is 3 ns, and hence the total hologram reconfiguration time depends mainly on the configuration of the control circuit. In our experiment, the PIN diodes are controlled in parallel by FPGA, the reconfiguration time is completely determined by the FPGA clock (CLK) rate, which is 100 MHz, corresponding to the time of each operation cycle of 10 ns. To change the PIN diode status in parallel, 3 operation cycles are needed after compiling. Thus the total hologram reconfiguration time is around 33ns.

4. Authors should comment if their demonstration scheme is scalable for optical frequencies.

Our Response:

Thank you very much for the suggestion. In fact, the reconfigurable metasurface hologram could also be realized at higher frequencies, to which a key issue is the construction of 1-bit phase-modulated light spatial mask. We note that several developed schemes of switchable metamaterial could be used for this purpose, for instance, the four-color spatial light modulator in the terahertz frequency [1], the switchable VO₂ in the terahertz [2] and near-infrared [3] frequencies, and Ge₂Sb₂Te₅ in the near-infrared frequency [4-5], etc. By incorporating phase changing materials into appropriate nanoantenna design, it is possible to extend our concept of dynamic metasurface holography to higher frequencies through dynamically modifying the phase of scattering from each individual antennas, while maintaining the scattering magnitude.

[1] D. Shrekenhamer, J. Montoya, S. Krishna, and W. J. Padilla, "Four-color metamaterial absorber THz spatial light modulator," *Adv. Optical Mater.*, 1, 905-909, 2013

[2] D. Wang *et al.*, "Switchable ultrathin quarter-wave plate in terahertz using active phase-change metasurface," *Sci. Rep.* 5, 15020, Oct. 2015.

[3] M. J. Dicken *et al.*, "Frequency tunable near-infrared metamaterials based on VO₂ phase transition," *Opt. Express* 17, 18330, Sep. 2009.

[4] B. Gholipour, J. Zhang, K. F. MacDonald, D. W. Hewak, and N. I. Zheludev, "An all-optical, non-volatile, bidirectional, phase-change meta-switch," *Adv. Mater.* 25, 3050-3054, Jun. 2013.

[5] Q. Wang *et al.*, "Optically reconfigurable metasurfaces and photonic devices based on phase change materials," *Nat. Photonics* 10, 60-65, Dec. 2015.

This discussion has been added to the Conclusion in the revised manuscript.

5. The paper mentions no applications of the demonstrated microwave hologram. It should elaborate further on how microwave holograms could be useful.

Our Response:

We thank the reviewer very much for this suggestion. The microwave hologram may be used for revolutionarily advancing the computational imager. Additionally, another very important application may be for the military purpose, e.g., reproducing the digital electromagnetic environment. Since arbitrary and reconfigurable microwave holography is achievable as demonstrated in our work, the full control of wave is readily at hand, enabling different ways of manipulating beams. Then a plethora of applications can be readily yielded and anticipated.

We have added the possible applications at the end of the second paragraph, Page 2, in the revised manuscript: “Thus, it is of great importance and interest to design metasurface holograms with dynamic capabilities for various applications such as advancing revolutionarily the computational imager^{33,34}, wireless communication, and reproducing the digital EM environments.”

Reviewers' comments:

Reviewer #1 (Remarks to the Author):

The authors have addressed most of the comments raised in the referees. However, some points are still needed to be clarified in particular articles 2 and 3 of the first referee.

The issue of the upscaling to optical frequencies (raised by both referees) has not been addressed properly possibly due to misunderstanding of the authors regarding the actual question. The question refers to the possibility of using the actual physical approach presented by the authors (i.e. introducing an electrically tunable capacitor or load) into the scattering element, not to the general ability to modify the phase response of metasurfaces by changing the resonance frequencies of the individual elements (which is basically obvious). The authors should elaborate on their ideas about the practical ability to control individual pixels (which is far from being trivial or simple using PCM such as VO₂ or GST).

Regarding the efficiency (article 3), I would expect that the first two bullets in the list (quantization loss and illuminating beam shape) could be taken into account theoretically in a similar manner to that done in Nano Lett. 14, 2485 (2014) for optical frequencies. By quantifying the impact of these effects (which can be done theoretically) the authors could obtain a more realistic upper bound on their expected theoretical efficiency.

Assuming these points are properly addressed I will be more than happy to recommend publication of the manuscript in Nature Communications.

Reviewer #2 (Remarks to the Author):

The revised manuscript is vastly improved. The title is more focused towards Electromagnetic holograms. The paper is now focused towards the achieved results and the application of the hologram is further elaborated. All comments of the previous review have been addressed appropriately.

The paper could be considered for publication in nature communication.

Re: Manuscript # NCOMMS-16-28675A “Electromagnetic Reprogrammable Coding Metasurface Holograms,” by L. Li et al.

Responses to Reviewer’s Comments

To Reviewer #1

Comments: The issue of the upscaling to optical frequencies (raised by both referees) has not been addressed properly possibly due to misunderstanding of the authors regarding the actual question. The question refers to the possibility of using the actual physical approach presented by the authors (i.e. introducing an electrically tunable capacitor or load) into the scattering element, not to the general ability to modify the phase response of metasurfaces by changing the resonance frequencies of the individual elements (which is basically obvious).

Response:

We apologize for misunderstanding the question in our earlier reply.

The proposed approach could be readily up-scaled to the THz frequencies, infrared, and beyond, as discussed below.

1) Extension to the THz regime

The metasurface unit can be down-scaled to the size of $300 \times 300 \mu\text{m}^2$ with slight modifications, specifically, a pair of gold pads with a separation of $30 \mu\text{m}$ atop a quartz substrate, as shown in Fig. R1(a). The ON/OFF states of metasurface unit can be controlled by using a commercial THz Schottky diode (AP1/G2/0P95), which is available at:

<http://www.teratechcomponents.com/products-and-services/>.

The parameters of this Schottky diode used in our simulations are listed in Table R1.

Table 1. Parameters of Schottky Diode Junction

Series Resistance	Saturation Current	Turn-on Voltage	Zero-bias Capacitance	Cut-off Frequency
13.67 Ω	0.98fA	0.62V	1.42fF	8THz

We have conducted CST-based numerical simulations to calculate the phase and amplitude responses of the proposed structure when the oxide is “ON” and “OFF”, respectively, as illustrated in Fig. R1(c) and (d), respectively. Nearly opposite phases are observed for the two states of the diode, showing the “0” and “1” digital states in the THz frequency.

Figure R1: (a) The THz metasurface unit, (b) the map of THz Schottky Diode from <http://www.teratechcomponents.com/products-and-services/>, (c) Phase responses and (d) amplitude response of the modified structure in the THz frequency when the oxide is “ON” and “OFF”, respectively.

2) Extension to the infrared-and-visible regime

Similar to above, the metasurface unit in the infrared-and-visible regime can be built, whose 0/1 states can be controlled by incorporating vanadium dioxide (VO₂) as the switching material^{R1}. A CST-based simulation is shown in Fig. R2, from which we can see that despite relatively low efficiency, nearly opposite phases are observed for the two states of the oxide, showing the “0” and “1” digital states in the infrared frequency. In practice, the phase transition of VO₂ within each unit cell may be individually controlled by optical pumping.

FIG. R2. (a) Infrared metasurface unit, (b) phase response and (c) amplitude response of the modified structure when its 0/1 states are controlled by VO₂ diode respectively. Here, simulation parameters of VO₂ dioxide are taken from Ref. [R1, R2].

[R1]Ghanekar A., et al., High-rectification near-field thermal diode using phase change periodic nanostructure, *Appl. Phys. Lett.*, **109**, 123106, 2016

[R2]Barker A. S., et al., Infrared optical properties of vanadium dioxide above and below the transition temperature, *Phys. Rev. Lett.*, **17**, 1286-1289, 1966

Comments: Regarding the efficiency, I would expect that the first two bullets in the list (quantization loss and illuminating beam shape) could be taken into account theoretically in a similar manner to that done in Nano Lett. 14, 2485 (2014) for optical frequencies.

Response:

We thank the reviewer for the suggestion and for providing the important reference.

The efficiency reduction due to the illumination beam shape: Assuming that f is the field at the hologram generated by an ideal plane-wave illumination, and g the actual illumination beam profile at the hologram plane, the real holographic image denoted by h can be approximated as the two-dimensional convolution of f and $\mathcal{F}\{g\}$, i.e., $h \approx f \otimes \mathcal{F}\{g\}$, where $\mathcal{F}\{g\}$ denotes the two-dimensional Fourier transformation of g . Apparently, the convolution kernel $\mathcal{F}\{g\}$ will lead to a blurred holographic image, which is responsible for the efficiency reduction.

The efficiency reduction due to the quantization: Following the suggested reference (Nano Lett. 14, 2485, 2014), we have performed the analysis of efficiency versus the number of

quantization level, as illustrated in Figure R3. We observe that the choice of two quantization levels will give the maximum attainable efficiency to nearly 70% of the optimal design efficiency.

Figure R3 The quantization efficiency penalty relative to the continuous phase solution, as a function of the number of quantization levels of the phase.

The above discussions have been added to the revised version of the manuscript and Supplementary Information, as marked in yellow colors.

REVIEWERS' COMMENTS:

Reviewer #1 (Remarks to the Author):

The authors have addressed all the comments and requests made by the reviewers. Particularly they show viable approaches for extending their proof of concept to the THz and optical spectral bands. The analysis of the impact of the phase quantization and beam divergence on the overall efficiency of the holograms provides a clear understanding of the efficiency penalty and reconciles between the theoretical and experimental results.

In view of the above I am pleased to recommend this paper for publication in Nature communications.

REVIEWERS' COMMENTS:

Reviewer #1:

Reviewer's comment:

The authors have addressed all the comments and requests made by the reviewers. Particularly they show viable approaches for extending their proof of concept to the THz and optical spectral bands. The analysis of the impact of the phase quantization and beam divergence on the overall efficiency of the holograms provides a clear understanding of the efficiency penalty and reconciles between the theoretical and experimental results. In view of the above I am pleased to recommend this paper for publication in Nature communications.

Our reply:

We thank the reviewer for recommending our paper for publication.